# Solving Inverse Problems in Protein Space Using Diffusion-Based Priors

## Abstract

The interaction of a protein with its environment can be understood and controlled via its 3D structure. Experimental methods for protein structure determination, such as X-ray crystallography or cryogenic electron microscopy, shed light on biological processes but introduce challenging inverse problems. Learning-based approaches have emerged as accurate and efficient methods to solve these inverse problems for 3D structure determination, but are specialized for a predefined type of measurement. Here, we introduce a versatile framework to turn biophysical measurements, such as cryo-EM density maps, into 3D atomic models. Our method combines a physics-based forward model of the measurement process with a pretrained generative model providing a task-agnostic, data-driven prior. Our method outperforms posterior sampling baselines on linear and nonlinear inverse problems. In particular, it is the first diffusion-based method for refining atomic models from cryo-EM maps and building atomic models from sparse distance matrices.

## 1 Introduction

Experimental methods in structural biology such as X-ray crystallography, cryogenic electron microscopy (cryo-EM) and nuclear magnetic resonance (NMR) spectroscopy provide noisy and partial measurements from which the 3D structure of biomolecules can be inferred. This three-dimensional information is key for our understanding of the molecular machinery of living organisms, as well as for designing therapeutic compounds. However, turning experimental observations into reliable 3D structural models is a challenging computational task. For many years, reconstruction algorithms were based on Maximum-A-Posteriori (MAP) estimation and often resorted to hand-crafted priors to compensate for the ill-posedness of the problem. State-of-the-art algorithms for cryo-EM reconstruction (Scheres, 2012; Punjani et al., 2017) are instances of such "white-box" algorithms. These approaches sometimes provide estimates for the uncertainty of their answers but can only leverage explicitly defined regularizers and do not cope well with complex noise sources or missing data.

Recently, supervised-learning approaches have emerged as an alternative to the MAP framework and some of them established a new empirical state of the art for certain tasks, like model building (Jamali et al., 2024). Typically, these supervised learning methods view the reconstruction problem as a regression task where a mapping between experimental measurements and atomic models needs to be learned. Some of these, like ModelAngelo, can even combine experimental data with sequence information by leveraging a pretrained protein language model (Rives et al., 2021). However, these methods must be trained on *paired* data (i.e., must be given input–output pairs) and can only cope with a predefined type of input. If additional information is available in a format that the model was not trained on (e.g., structural information about a fragment of the protein), or if the distribution of input data shifts at inference time (e.g., if the noise level changes due to modifications in the experimental protocol), a new model needs to be trained to properly cope with the new data.

In the field of imaging, scenarios where an image or a 3D model must be inferred from corrupted and partial observations are known as "inverse problems". To overcome the ill-posedness of these problems, regularizers were heuristically defined to inject hand-crafted priors and turn Maximum Likelihood Estimation (MLE) problems into MAP problems. In a similar fashion, machine learning-based methods were recently shown to outperform hand-crafted algorithms for a wide variety of tasks: denoising (Zhang et al., 2017), inpainting (Xie et al., 2012), super-resolution (Lim et al.,

2017), deblurring (Nah et al., 2017), monocular depth estimation (Eigen et al., 2014), and camera calibration (Kendall et al., 2015), among others. These methods, however, are equally limited by their need for paired data and their poor performance in the eventuality of a distribution shift.

In contrast, the MAP approach does not require paired data and can leverage the knowledge of the physics behind the problem through the definition of a likelihood function. As for the prior, generative models were shown to be effective tools to inject data-driven priors into MAP problems, making inverse problems well-posed while circumventing the need for heuristic priors (Bora et al., 2017). Among these generative methods, diffusion models gained popularity due to their powerful capabilities in the unconditional generation of images (Dhariwal & Nichol, 2021), videos (Ho et al., 2022), and 3D assets (Po et al., 2023), and were recently leveraged to solve inverse problems in image space (Song et al., 2022; Chung et al., 2022a). The field of structural biology has also witnessed the application of diffusion models in protein structure modeling tasks (Watson et al., 2023; Abramson et al., 2024). The recently released generative model Chroma (Ingraham et al., 2023) stands out in part thanks to its "programmable" framework, i.e., its ability to be conditioned on external hard or soft constraints, but was never applied to structure determination problems like atomic model building.

Here, we introduce ADP-3D (Atomic Denoising Prior for 3D reconstruction), a framework to condition a diffusion model in protein space with any observations for which the measurement process can be physically modeled. Instead of using unadjusted Langevin dynamics for posterior sampling, our approach performs MAP estimation and leverages the data-driven prior learned by a diffusion model using the plug-n-play framework (Venkatakrishnan et al., 2013), (Zhu et al., 2023). We demonstrate that our method handles a variety of external information: cryo-EM density maps, amino acid sequence, partial 3D structure, and pairwise distances between amino acid residues, to refine a complete 3D atomic model of the protein. We show that our method outperforms a posterior sampling baseline in average accuracy and, given a cryo-EM density map, can accurately refine incomplete atomic models provided by ModelAngelo. ADP-3D can leverage any protein diffusion model as a prior, which we demonstrate by showing results obtained with Chroma (Ingraham et al., 2023) and RFdiffusion (Watson et al., 2023). We therefore make the following contributions:

- We introduce a versatile framework, inspired by plug-n-play, to solve inverse problems in protein space with a pretrained diffusion model as a learned prior;
- We outperform an existing posterior sampling method at reconstructing full protein structures from partial structures;
- We show that a protein diffusion model can be guided to perform atomic model refinement in simulated and experimental cryo-EM density maps;
- We show that a protein diffusion model can be conditioned on a sparse distance matrix.

## 2 RELATED WORK

**Protein Diffusion Models.** Considerable progress has been made in leveraging diffusion models for protein structure generation. While the first models could sample distance matrices (Lee et al., 2022), they were later improved to directly sample backbone structures represented by 3D point clouds (Anand & Achim, 2022; Trippe et al., 2022), backbone internal coordinates (Wu et al., 2024), 3D "frames" (Yim et al., 2023). Recent methods are able to directly sample all-atom structures, including side chains (Chu et al., 2024). In RFdiffusion, Watson et al. (2023) experimentally designed the generated proteins and structurally validated them with cryo-EM. In Chroma, Ingraham et al. (2023) introduced a "conditioning" framework to generate proteins with desired properties (e.g., substructure motifs, symmetries), but this framework was never used to enable protein structure determination from exprimental measurements. Recently, AlphaFold 3 (Abramson et al., 2024) showed that a diffusion model operating on raw atom coordinates could be used as a tool to improve protein structure prediction. As generative models for proteins keep improving, leveraging them in the most impactful way becomes an increasingly important matter.

Here, we introduce a framework to efficiently condition a pretrained protein diffusion model and demonstrate the possibility of using cryo-EM maps as conditioning information. Most of our experiments are conducted using Chroma as a prior and we provide additional results using RFdiffusion in the supplements.

**Diffusion-Based Posterior Sampling in Image Space.** An *inverse problem* in image space can be defined by $\mathbf{y} = \Gamma(\mathbf{x}) + \eta$ where $\mathbf{x}$ is an unknown image, $\mathbf{y}$ a measurement, $\Gamma$ a known operator and $\eta$ a noise vector of known distribution, potentially signal-dependent. The goal of posterior sampling is to sample $\mathbf{x}$ from the posterior $p(\mathbf{x}|\mathbf{y})$, the normalized product of the prior $p(\mathbf{x})$ and the likelihood $p(\mathbf{y}|\mathbf{x})$. Bora et al. (2017) showed that generative models could be leveraged to implicitly represent a data-learned prior and solve compressed sensing problems in image space. Motivated by the success of diffusion models at unconditional generation (Dhariwal & Nichol, 2021), several works showed that score-based and denoising models could be used to solve linear inverse problems like super-resolution, deblurring, inpainting and colorization (Li et al., 2022; Choi et al., 2021; Saharia et al., 2022; Kawar et al., 2022; Lugmayr et al., 2022; Zhu et al., 2023), leading to results of unprecedented quality. Other methods leveraged the score learned by a diffusion model to solve inverse problems in medical imaging (Song et al., 2021; Jalal et al., 2021; Chung & Ye, 2022; Chung et al., 2022c;b) and astronomy (Sun et al., 2023). Finally, recent methods went beyond the scope of linear problems and used diffusion-based posterior sampling on nonlinear problems like JPEG restoration (Song et al., 2022), phase retrieval and non-uniform deblurring (Chung et al., 2022a). We refer to Daras et al. (2024) for an in-depth survey of the methods leveraging diffusion models as priors in inverse problems. Taking inspiration from these methods, and in particular from DiffPIR (Zhu et al., 2023), we propose to leverage protein diffusion models to solve nonlinear inverse problems in protein space.

**Model Building Methods.** Cryogenic electron-microscopy (cryo-EM) provides an estimate of the 3D electron scattering potential (or density map) of a protein. The task of fitting an atomic model $\mathbf{x}$ into this 3D map $\mathbf{y}$ is called model building and can be seen as a nonlinear inverse problem in protein space (see 4.3).

Model building methods were first developed in X-Ray crystallography (Cowtan, 2006) and automated methods like Rosetta *de-novo* (Wang et al., 2015), PHENIX (Liebschner et al., 2019; Terwilliger et al., 2018) and MAINMAST (Terashi & Kihara, 2018) were later implemented for cryo-EM data. Although they constituted a milestone towards the automation of model building, obtained structures were often incomplete and needed refinement (Singharoy et al., 2016). Supervised learning techniques were applied to model building, relying on U-Net-based architectures (Si et al., 2020; Zhang et al., 2022; Pfab et al., 2021), or combining a 3D transformer with a Hidden Markov Model (Giri & Cheng, 2024). EMBuild (He et al., 2022) was the first method to make use of sequence information and ModelAngelo (Jamali et al., 2024) established a new state of the art for automated *de novo* model building. Trained on 3,715 experimental paired datapoints, ModelAngelo uses a GNN-based architecture and processes the sequence information with a pretrained language model (Rives et al., 2021). Although fully-supervised methods outperform previous approaches, they still provide incomplete atomic models and cannot use a type of input data it was not trained with as additional information.

Here, we propose a versatile framework to solve inverse problems in protein space, including atomic model refinement. Our approach can cope with auxiliary measurements for which the measurement process is known. Our framework allows any pretrained diffusion model to be plugged-in as a prior and can therefore take advantage of future developments in generative models without any task-specific retraining step.

## 3 BACKGROUND

### 3.1 DIFFUSION IN PROTEIN SPACE WITH CHROMA

In Chroma (Ingraham et al., 2023), the atomic structure of a protein of $N$ amino acid residues is represented by the 3D Cartesian coordinates $\mathbf{x} \in \mathbb{R}^{4N \times 3}$ of the four backbone heavy atoms (N, $C_\alpha$, C, O) in each residue, the amino acid sequence $\mathbf{s} \in \{1, ..., 20\}^N$, and the side chain torsion angles for each amino acid $\chi \in (-\pi, \pi]^{4N}$ (the conformation of the side chain can be factorized as up to four sequential rotations). The joint distribution over all-atom structures is factorized as

$$p(\mathbf{x}, \mathbf{s}, \chi) = p(\mathbf{x})p(\mathbf{s}|\mathbf{x})p(\chi|\mathbf{x}, \mathbf{s}). \tag{1}$$

The first factor on the right hand side, $p(\mathbf{x})$, is modeled as a diffusion process operating in the space of backbone structures $\mathbf{x}$. Given a structure $\mathbf{x}$ at diffusion time $t$, Chroma models the conditional distribution of the sequence $p_\theta(\mathbf{s}|\mathbf{x}, t)$ as a conditional random field and the conditional distribution of the side chain conformations $p_\theta(\chi|\mathbf{x}, \mathbf{s}, t)$ with an autoregressive model.

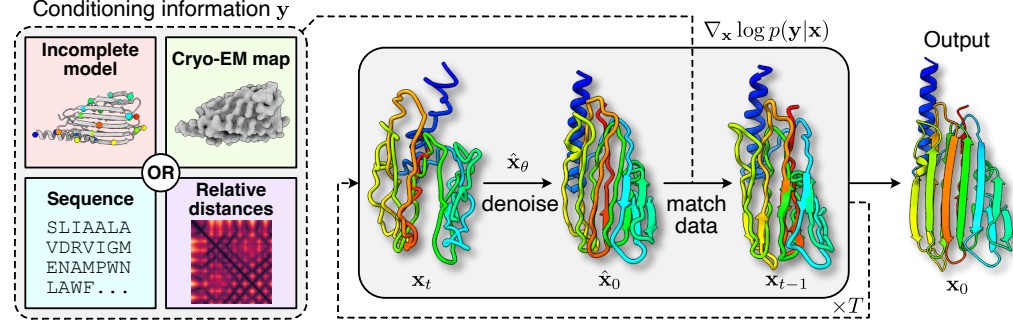

**Figure 1: Overview of ADP-3D.** Our method turns partial and noisy measurements (the "conditioning information") into a 3D structure by leveraging a pretrained diffusion model and physics-based models of the measurement processes. Starting from a random structure $\mathbf{x}_T$, our method iterates between a denoising step and a data-matching step. The denoiser comes from the pretrained diffusion model. The data-matching step aims at maximizing the likelihood of the measurements.

Adding isotropic Gaussian noise to a backbone structure would rapidly destroy simple biophysical patterns that proteins always follow (e.g., the scaling law of the radius of gyration with the number of residues). Instead, Chroma uses a non-isotropic noising process as an inductive bias to alleviate the need for the model to learn these patterns from the data. The correlation of the noise is defined in such a way that a few structural properties are statistically preserved throughout the noising process. Specifically, the forward diffusion process is defined by the variance-preserving stochastic process

$$\mathrm{d}\mathbf{x} = -\frac{1}{2}\mathbf{x}\beta_t\mathrm{d}t + \sqrt{\beta_t}\mathbf{R}\mathrm{d}\mathbf{w}, \tag{2}$$

where $\beta_t$ is a time-dependent noising schedule and $\mathrm{d}\mathbf{w}$ is a standard Wiener process of dimension $\mathbb{R}^{4N \times 3}$. The matrix $\mathbf{R} \in \mathbb{R}^{4N \times 4N}$ is fixed and defined explicitly based on statistical considerations regarding the structure of proteins (see Ingraham et al. (2023) and supplements). Starting from $\mathbf{x}_0$ at $t = 0$, a solution to this stochastic differential equation (SDE) at time $t$ is given by

$$\mathbf{x}_t \sim \mathcal{N}(\mathbf{x}; \alpha_t\mathbf{x}_0, \sigma_t^2\mathbf{R}\mathbf{R}^T), \tag{3}$$

where $\alpha_t = \exp\left(-\frac{1}{2}\int_0^t \beta_s\mathrm{d}s\right)$ and $\sigma_t = \sqrt{1 - \alpha_t^2}$.

New protein samples can be generated by sampling $\mathbf{x}_T$ from $\mathcal{N}(0, \mathbf{R}\mathbf{R}^T)$ and integrating the following reverse-time SDE over $t \in [T, 0]$ (Anderson, 1982):

$$\mathrm{d}\mathbf{x} = \left[-\frac{1}{2}\mathbf{x} - \mathbf{R}\mathbf{R}^T\nabla_{\mathbf{x}}\log p_t(\mathbf{x})\right]\beta_t\mathrm{d}t + \sqrt{\beta_t}\mathbf{R}d\bar{\mathbf{w}}, \tag{4}$$

where $d\bar{\mathbf{w}}$ is a reverse-time Wiener process. Following Tweedie's formula (Robbins, 1992), the score $\nabla_{\mathbf{x}}\log p_t(\mathbf{x})$ is an affine function of the time-dependent *optimal denoiser*, approximated by $\hat{\mathbf{x}}_\theta(\mathbf{x}, t)$:

$$\nabla_{\mathbf{x}}\log p_t(\mathbf{x}) = \frac{(\mathbf{R}\mathbf{R}^T)^{-1}}{1 - \alpha_t^2}(\alpha_t\mathbb{E}[\mathbf{x}_0|\mathbf{x}_t = \mathbf{x}] - \mathbf{x}), \quad \hat{\mathbf{x}}_\theta(\mathbf{x}, t) \approx \mathbb{E}[\mathbf{x}_0|\mathbf{x}_t = \mathbf{x}]. \tag{5}$$

## 3.2 HALF QUADRATIC SPLITTING AND PLUG-N-PLAY FRAMEWORK

An objective function of the form $f(\mathbf{x}) + g(\mathbf{x})$ can be efficiently minimized over $\mathbf{x}$ using a variable splitting algorithm like Half Quadratic Splitting (HQS) (Geman & Yang, 1995). By introducing an auxiliary variable $\tilde{\mathbf{x}}$, the HQS method relies on iteratively solving two subproblems:

$$\begin{aligned}
\tilde{\mathbf{x}}_k &= \mathrm{prox}_{g,\gamma}(\mathbf{x}_k) = \arg\min_{\tilde{\mathbf{x}}} \ g(\tilde{\mathbf{x}}) + \frac{\gamma}{2}\|\tilde{\mathbf{x}} - \mathbf{x}_k\|_2^2, \\
\mathbf{x}_{k+1} &= \mathrm{prox}_{f,\gamma}(\tilde{\mathbf{x}}_k) = \arg\min_{\mathbf{x}} \ f(\mathbf{x}) + \frac{\gamma}{2}\|\mathbf{x} - \tilde{\mathbf{x}}_k\|_2^2,
\end{aligned} \tag{6}$$

where prox are called "proximal operators" and $\gamma > 0$ is a user-defined proximal parameter.

If $f$ represents a negative log-likelihood over $\mathbf{x}$ and $g$ represents a negative log-prior, the above problem defines a Maximum-A-Posterior (MAP) problem. The key idea of the plug-and-play framework (Venkatakrishnan et al., 2013) is to notice that the first minimization problem in equation 6 is

exactly a Gaussian denoising problem at noise level $\sigma = \sqrt{1/\gamma}$ with the prior $\exp(-g(\mathbf{x}))$ in $\mathbf{x}$-space. This means that any Gaussian denoiser can be used to "plug in" a prior into a MAP problem.

Once a diffusion model has been trained, it provides a deterministic Gaussian denoiser for various noise levels, as described in equation 5. As recently shown in Zhu et al. (2023), this optimal denoiser can be used in the plug-n-play framework to solve MAP problems in image space. Here, we propose to apply this idea to inverse problems in protein space, leveraging a pretrained diffusion model.

## 4 METHODS

In this section, we formulate our method, ADP-3D (Atomic Denoising Prior for 3D reconstruction), as a MAP estimation method in protein space and explain how the plug-n-play framework can be used to leverage the prior learned by a pretrained diffusion model. The method is described visually in Figure 1. We then introduce our preconditioning strategy in the case of linear problems. Finally, we describe and model the measurement process in cryogenic electron microscopy. ADP-3D is described with pseudo-code in Algorithm 1.

### 4.1 GENERAL APPROACH

Given a set of independent measurements $\mathbf{Y} = \{\mathbf{y}_i\}_{i=1}^n$ made from the same unknown protein, our goal is to find a Maximum-A-Posteriori (MAP) estimate of the backbone structure $\mathbf{x}^*$. Following Bayes' rule,

$$\mathbf{x}^* = \arg\max_{\mathbf{x}} \{p_0(\mathbf{x}|\mathbf{Y})\} = \arg\min_{\mathbf{x}} \Big\{ \underbrace{- \sum_{i=1}^n \log p_0(\mathbf{y}_i|\mathbf{x})}_{f(\mathbf{x})} \underbrace{- \log p_0(\mathbf{x})}_{g(\mathbf{x})} \Big\}. \quad (7)$$

While most of previous works leveraging a diffusion model for inverse problems aim at sampling from the posterior distribution $p(\mathbf{x}|\mathbf{Y})$, we are interested here in scenarios where the measurements convey enough information to make the MAP estimate unique and well-defined.

We take inspiration from the plug-and-play framework (Venkatakrishnan et al., 2013) to efficiently solve equation 7. We propose to use the optimal denoiser $\hat{\mathbf{x}}_\theta(\mathbf{x}, t)$ of a pretrained diffusion model to solve the first subproblem in equation 6. Framing the optimization loop in the whitened space of $\mathbf{z} = \mathbf{R}^{-1}\mathbf{x}$, which provides more stable results, our general optimization algorithm can be summarized in three steps:

$$\tilde{\mathbf{z}}_0 = \mathbf{R}^{-1}\hat{\mathbf{x}}_\theta(\mathbf{R}\mathbf{z}_t, t) \qquad\qquad \text{Denoise at level } t,$$

$$\hat{\mathbf{z}}_0 = \arg\min_{\mathbf{z}} \frac{\gamma}{2} \|\mathbf{z} - \tilde{\mathbf{z}}_0\|_2^2 - \sum_{i=1}^n \log p_0(\mathbf{y}_i|\mathbf{z}) \qquad \text{Maximize likelihood,}$$

$$\mathbf{z}_{t-1} \sim \mathcal{N}(\alpha_{t-1}\hat{\mathbf{z}}_0, \sigma_{t-1}^2) \qquad\qquad \text{Add noise at level } t-1.$$

Here, no specific assumptions have been made on the likelihood term and this framework could hypothetically be applied on any set of measurements for which we have a physics-based model of the measurement process. Since the second step is not tractable in most cases, we replace the explicit minimization with a gradient step with momentum from the iterate $\tilde{\mathbf{z}}_0$. This step can be implemented efficiently using automatic differentiation. The gradient of $\|\mathbf{z} - \hat{\mathbf{z}}_0\|_2^2$ w.r.t $\mathbf{z}$ in $\hat{\mathbf{z}}_0$ being null, the method does not depend on the choice $\gamma$.

### 4.2 PRECONDITIONING FOR LINEAR MEASUREMENTS

We consider the case where the measurement process is linear:

$$\mathbf{y} = \mathbf{A}\mathbf{x}_0 + \eta = \mathbf{A}\mathbf{R}\mathbf{z}_0 + \eta, \quad \eta \sim \mathcal{N}(0, \boldsymbol{\Sigma} \in \mathbb{R}^{m \times m}), \quad (8)$$

with $\mathbf{y} \in \mathbb{R}^m$ and $\mathbf{A} \in \mathbb{R}^{m \times 4N}$ being a known measurement matrix of rank $m$. In this case, the log-likelihood term is a quadratic function:

$$\log p_0(\mathbf{y}|\mathbf{z}) = -\frac{1}{2}\|\mathbf{A}\mathbf{R}\mathbf{z} - \mathbf{y}\|_{\boldsymbol{\Sigma}^{-1}}^2 + C, \text{ where } \|\mathbf{x}\|_{\boldsymbol{\Sigma}^{-1}}^2 = \mathbf{x}^T\boldsymbol{\Sigma}^{-1}\mathbf{x}, \quad (9)$$

---

**Algorithm 1** ADP-3D (Atomic Denoising Prior for 3D reconstruction)

---

**Inputs**: log-likelihood functions $\{f_i : (\mathbf{y}, \mathbf{z}) \mapsto \log p(\mathbf{y}_i = \mathbf{y}|\mathbf{z})\}_{i=1}^n$, measurements $\{\mathbf{y}_i\}$.
**Diffusion model**: correlation matrix $\mathbf{R}$, denoiser $\hat{\mathbf{x}}_\theta(\mathbf{x}, t)$, schedule $\{\alpha_t, \sigma_t\}_{t=1}^T$.
**Optimization parameters**: learning rates $\{\lambda_i\}$, momenta $\{\rho_i\}$.
**Initialization**: $\mathbf{z}_T \leftarrow \mathcal{N}(0, \mathbf{I}), \quad \forall i, \mathbf{v}_i = 0$
**for** $t = T, \dots, 1$ **do**
$\quad \tilde{\mathbf{z}}_0 \leftarrow \mathbf{R}^{-1}\hat{\mathbf{x}}_\theta(\mathbf{R}\mathbf{z}_t, t)$ ⟶ Denoise at level $t$
$\quad \forall i, \mathbf{v}_i = \rho_i \mathbf{v}_i + \lambda_i \nabla_{\mathbf{z}} f_i(\mathbf{y}_i, \mathbf{z})|_{\mathbf{z}=\tilde{\mathbf{z}}_0}$ ⟶ Accumulate gradient of log-likelihood
$\quad \hat{\mathbf{z}}_0 \leftarrow \tilde{\mathbf{z}}_0 + \sum_i \mathbf{v}_i$ ⟶ Take a step to maximize likelihood
$\quad \mathbf{z}_{t-1} \sim \mathcal{N}(\alpha_{t-1}\hat{\mathbf{z}}_0, \sigma_{t-1}^2)$ ⟶ Add noise at level $t-1$
**end for**
**return** $\mathbf{x}_0 = \mathbf{R}\mathbf{z}_0$

---

and $C$ does not depend on $\mathbf{z}$. As shown in the supplements, the condition number of $\mathbf{R}$ (i.e., the ratio between its largest and smallest singular values) grows as a power function of the number of residues. For typical proteins ($N \geq 100$), this condition number exceeds 100, making the maximization of the above term an ill-conditioned problem. In order to make gradient-based optimization more efficient, we propose to *precondition* the problem by precomputing a singular value decomposition $\mathbf{AR} = \mathbf{USV}^T$ and to set $\mathbf{\Sigma} = \sigma^2 \mathbf{USS}^T\mathbf{U}^T$. Note that this is equivalent to modeling the measurement process as $\mathbf{y} = \mathbf{AR}(\mathbf{z} + \tilde{\eta})$ with $\tilde{\eta} \sim \mathcal{N}(0, \sigma^2)$. In other words, we assume that the noise $\eta$ preserves the simple patterns in proteins, which is a reasonable hypothesis if, for example, $\mathbf{y}$ is an incomplete atomic model obtained by an upstream reconstruction algorithm that leverages prior knowledge on protein structures. The log-likelihood then becomes

$$\log p_0(\mathbf{y}|\mathbf{z}) = -\frac{1}{2\sigma^2} \left\| \begin{pmatrix} \mathbf{I}_m & 0 \\ 0 & 0 \end{pmatrix} \mathbf{V}^T\mathbf{z} - \mathbf{S}^+\mathbf{U}^T\mathbf{y} \right\|_2^2 + C. \tag{10}$$

The maximization of this term is a well-posed problem that gradient ascent with momentum efficiently solves (see supplementary analyses). In equation 10, $\mathbf{S}^+$ denotes the pseudo-inverse of $\mathbf{S}$.

### 4.3 Application to Atomic Model Building

**Measurement Model in Cryo-EM.** In single particle cryo-EM, a purified solution of a target protein is flash-frozen and imaged with a transmission electron microscope, providing thousands to millions of randomly oriented 2D projection images of the protein's electron scattering potential. Reconstruction algorithms process these images and infer a 3D *density map* of the protein. Given a protein $(\mathbf{x}, \mathbf{s}, \chi)$, its density map is well approximated by (De Graef, 2003)

$$\mathbf{y} = \mathcal{B}(\Gamma(\mathbf{x}, \mathbf{s}, \chi)) + \eta \in \mathbb{R}^{D \times D \times D}, \tag{11}$$

where $\Gamma$ is an operator that places a sum of 5 isotropic Gaussians centered on each heavy atom. The amplitudes and standard deviations of these Gaussians, known as "form factors", are tabulated (Hahn et al., 1983) and depend on the chemical element they are centered on. $\mathcal{B}$ represents the effect of "B-factors" (Kaur et al., 2021) and can be viewed as a spatially dependent blurring kernel modelling molecular motions and/or signal damping by the transfer function of the electron microscope. $\eta$ models isotropic Gaussian noise of variance $\sigma^2$. This measurement model leads to the following log-likelihood:

$$\log p_0(\mathbf{y}|\mathbf{x}, \mathbf{s}, \chi) = -\frac{1}{2\sigma^2} \|\mathbf{y} - \mathcal{B}(\Gamma(\mathbf{x}, \mathbf{s}, \chi))\|_2^2 + C. \tag{12}$$

**Likelihood Terms in Model Refinement.** We consider a 3D density map $\mathbf{y}$ provided by an upstream reconstruction method and an incomplete backbone structure $\bar{\mathbf{x}} \in \mathbb{R}^m$ ($m \leq 4N$) provided by an upstream model building algorithm (e.g., ModelAngelo (Jamali et al., 2024)). Sequencing a protein is now a routine process (De Hoffmann & Stroobant, 2007) and we therefore consider the sequence $\mathbf{s}$ as an additional source of information. The side chain angles $\chi$ are, however, unknown.

The log-likelihood of our measurements for a given backbone structure $\mathbf{x}$ can be decomposed as

$$\log p_0(\mathbf{y}, \mathbf{s}, \bar{\mathbf{x}}|\mathbf{x}) = \log p_0(\bar{\mathbf{x}}|\mathbf{x}) + p_0(\mathbf{y}, \mathbf{s}|\mathbf{x}) = \log p_0(\bar{\mathbf{x}}|\mathbf{x}) + \log p_0(\mathbf{y}|\mathbf{x}, \mathbf{s}) + \log p_0(\mathbf{s}|\mathbf{x}). \tag{13}$$

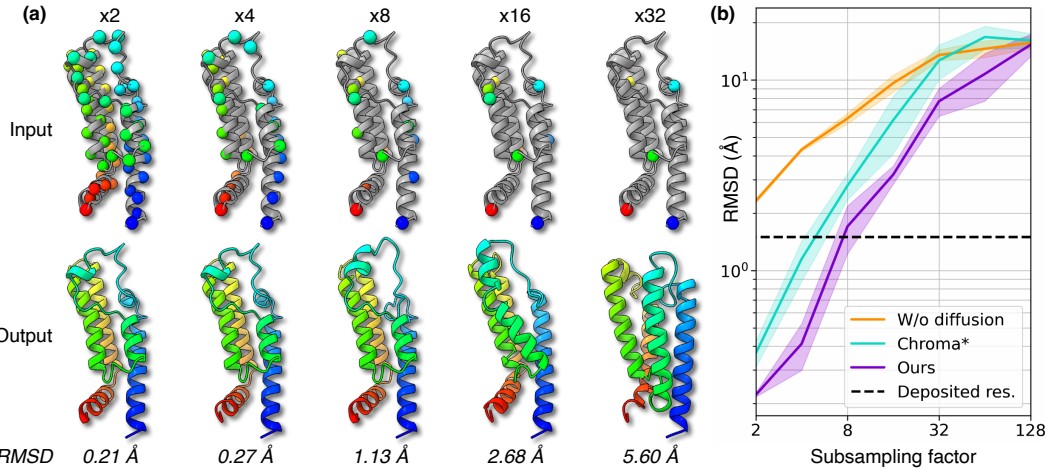

**Figure 2: Structure Completion.** Results on the ATAD2 protein (PDB: 7qum, 130 residues). (a) Qualitative results. The input structure is a subsampled version of the target structure (subsampling factor in the top row). In the "input" row, we show the target (unknown) in gray and the locations of the known alpha carbons in colors. We report the lowest RMSD over 8 runs. (b) RMSD vs. subsampling factor. Our method is compared to Chroma conditioned with the `SubstructureConditioner`. The importance of the diffusion-based prior is shown. We report the mean RMSD ($\pm 1$ std) over 8 runs. The experimental (deposited) resolution is indicated with a dashed line.

On the right-hand side, the last term can be approximated using the learned conditional distribution $p_\theta(\mathbf{s}|\mathbf{x})$. We model $\bar{\mathbf{x}} = \mathbf{Mx} + \eta$ so that the first term can be handled by the preconditioning procedure described in the previous section. Finally, the middle term involves the marginalization of $p_0(\mathbf{y}|\mathbf{x}, \mathbf{s}, \chi)$ over $\chi$. This marginalization is not tractable but equation 12 provides a lower bound:

$$\log p_0(\mathbf{y}|\mathbf{x}, \mathbf{s}) \geq \mathbb{E}_{\chi \sim p_0(\chi|\mathbf{x}, \mathbf{s})} \left[\log p_0(\mathbf{y}|\mathbf{x}, \mathbf{s}, \chi)\right] \approx \mathbb{E}_{\chi \sim p_\theta(\chi|\mathbf{x}, \mathbf{s})} \left[\log p_0(\mathbf{y}|\mathbf{x}, \mathbf{s}, \chi)\right], \quad (14)$$

using Jensen's inequality. The expectation is approximated by Monte Carlo sampling and gradients of $\chi$ with respect to $\mathbf{x}$ are computed by automatic differentiation through the autoregressive sampler of $\chi$, following the "reparameterization trick" (Kingma, 2013).

## 5 EXPERIMENTS

**Experimental Setup.** Our main results are obtained using the publicly released version of Chroma[1] (Ingraham et al., 2023). We provide additional results with the publicly released version of RFdiffusion[2] (Watson et al., 2023) in the supplements. We run all our experiments using structures of proteins downloaded from the Protein Data Bank (PDB) (Burley et al., 2021). In order to select proteins that do not belong to the training dataset of Chroma, here we only consider structures that were released after 2022-03-20 (Chroma was trained on a filtered version of the PDB queried on that date). We provide additional results on structures taken from the CASP15 dataset in the supplements. For the proteins that are not fully modeled on the PDB, we mask out the residues with ground truth coordinates before computing the Root Mean Square Deviation (RMSD). In each experiment, we run 8 replicas in parallel on a single NVIDIA A100 GPU. Further details about each target structure are provided in the supplements.

**Structure Completion.** Given an incomplete atomic model of a protein, our first task is to predict the coordinates of all heavy atoms in the backbone. This first task is designed as a toy problem, with no immediate application to real data, to validate and evaluate our method. The forward measurement process can be modeled as $\mathbf{y} = \mathbf{Mx}$ where $\mathbf{M} \in \{0, 1\}^{(4N/k) \times 4N}$ is a *masking* matrix ($\mathbf{M1} = \mathbf{1}$) and $k$ is the *subsampling factor*. We consider the case where, for each residue, the location of all 4 heavy atoms on the backbone (N, $C_\alpha$, C, O) is either known or unknown. Residues of known locations are regularly spaced along the backbone every $k$ residues. We compare our results to the baseline Chroma conditioned with a `SubstructureConditioner` (Ingraham et al., 2023). This baseline samples from the posterior probability $p(\mathbf{x}|\mathbf{y})$ using unadjusted Langevin dynamics. We use 1000 diffusion steps for our method and the baseline.

---

[1] https://github.com/generatebio/chroma
[2] https://github.com/RosettaCommons/RFdiffusion

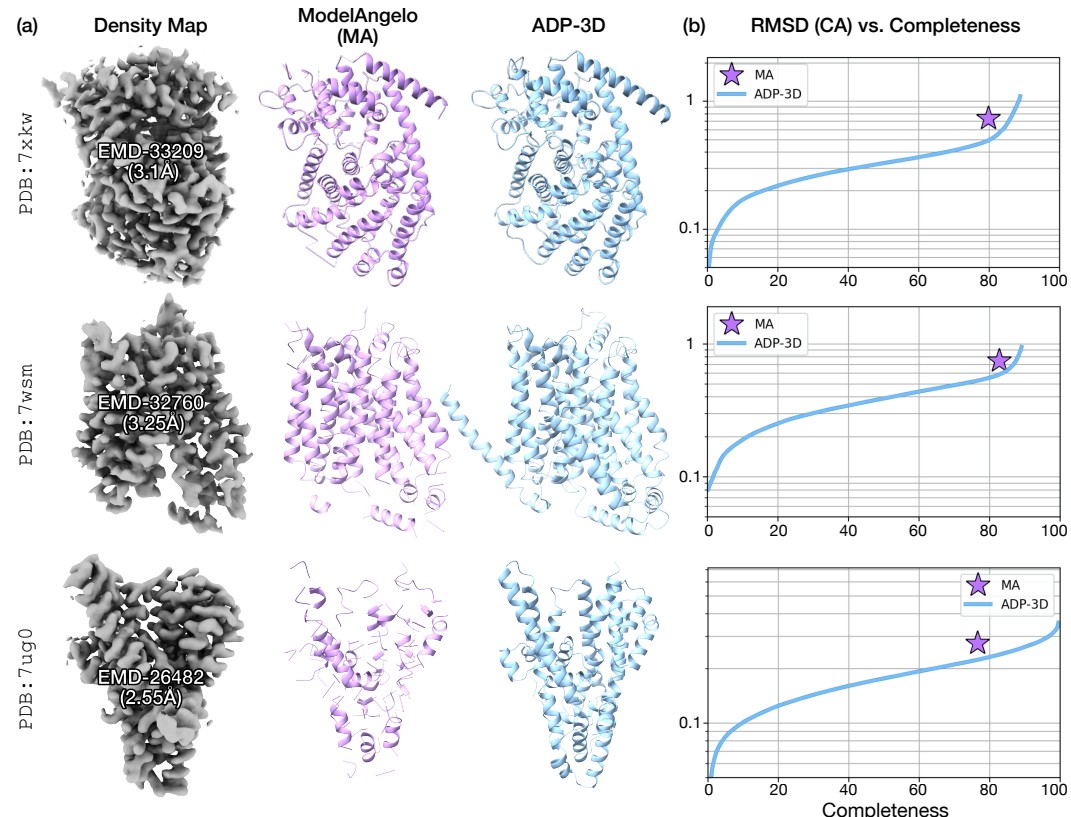

**Figure 3: Atomic Model Refinement.** (a) Experimental density map, ModelAngelo's incomplete model and ADP-3D's atomic model. (b) RMSD of alpha carbons vs. completeness (number of predicted residues / total number of residues) with ModelAngelo (MA) and our method. For EMD-26482 (third row), we remove uniformly sampled residues from the ModelAngelo model until reaching sub-80% of completeness. The RMSD is computed with respect to the deposited structure on the PDB.

In Figure 2, we show our results on ATAD2 (PDB:7qum) (Davison et al., 2022; Bamborough et al., 2016), a cancer-associated protein of 130 residues. The protein was resolved at a resolution of 1.5 Å using X-ray crystallography. Our method recovers the target structure without loss of information (RMSD < 1.5 Å) for subsampling factors of 2, 4 and 8. Fig. 2.b shows that our method outperforms the baseline and highlights the importance of the diffusion-based prior. When the subsampling factor is large ($\geq 32$), the reconstruction accuracy decreases but the method inpaints unknown regions with realistic secondary structures (see quantitative evaluation in the supplementary). Note that making the conditioning information sparser (increasing the subsampling factor) tends to close the gap between our method (MAP estimation) and the baseline (posterior sampling).

**Atomic Model Refinement.** Next, we evaluate our method on the model refinement task. We use experimental cryo-EM maps of single-chain structures: EMD-33209 (density map of PDB:7xkw Ye et al. (2024)), EMD-32760 (density map of PDB:7wsm Yuan et al. (2022)), EMD-26482 (density map of PDB:7ug0 Huang et al. (2023)). We directly use the publicly available versions of EMD-33209 and EMD-32760, and run ModelAngelo (Jamali et al., 2024) with its default parameters. For EMD-26482, the deposited map is a trimeric version of PDB:7ug0. We use the volume zone tool of ChimeraX to select and keep the regions of the density map within 3 Å of the deposited atomic model. We then run ModelAngelo using its default parameters. All the incomplete models provided by ModelAngelo are cleaned by removing the residues for which the $C\alpha$ atom is not located within $3.8 \pm 0.3$ Å of both neighboring $C\alpha$ atoms. For the model obtained from EMD-26482, we also randomly remove uniformly sampled residues in the incomplete model, such that the completeness gets below 80%. We provide ModelAngelo's output (an incomplete model) to our method, along with the density map and the sequence. To evaluate our method, we report the RMSD of the predicted structure for the $X\%$ most well-resolved alpha carbons (compared to the deposited structure), for $X \in [0, 100]$ ($X$ is called the "completeness").

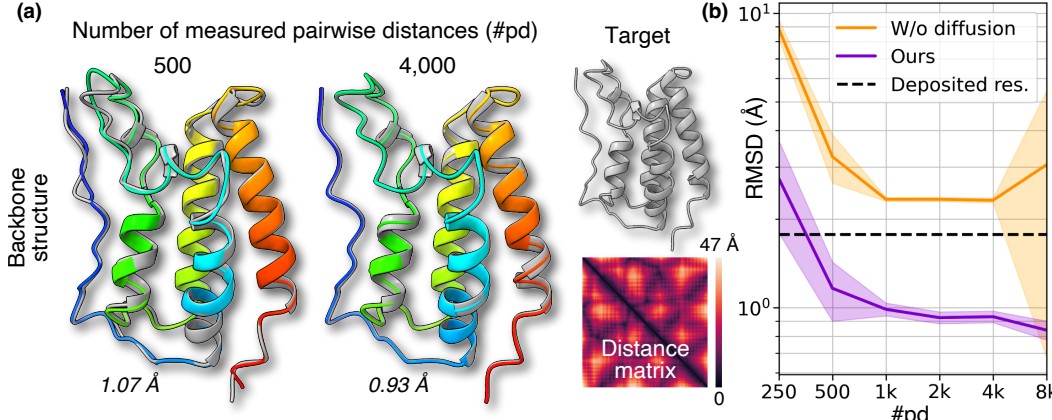

**Figure 4: Pairwise Distances to Structures.** Results on BRD4 (PDB:7r5b, 127 residues) (a) Qualitative results. The reconstructed structures are shown in colors, depending on the number of known pairwise distances. We report the lowest RMSD over 8 runs. The target structure is shown in gray along with its pairwise distance matrix. (b) RMSD vs. number of known pairwise distances. Each experiment is ran 10 times with randomly sampled distances. We report the mean of the lowest RMSD obtained over 8 replicas ($\pm 1$ std). The plot demonstrates the importance of the diffusion model. The experimental (deposited) resolution is indicated with a dashed line.

We show qualitative and quantitative results in Figure 3. For all structures, ADP-3D improves the accuracy of the ModelAngelo model for the same level of completeness. Note that the RMSD can only be computed up to the completeness level of the deposited structure. We provide additional results in the supplements and investigate on the influence of the resolution of the input density map and we perform an ablation study on the conditioning information.

**Pairwise Distances to Structure**  Finally, we assume we are given a set of pairwise distances between alpha carbons and we use our method to predict a full 3D structure. This task is a simplification of the reconstruction problem in paramagnetic NMR spectroscopy, where one can obtain information about the relative distances and orientations between pairs of atoms via the nuclear Overhauser effect and sparse paramagnetic restraints, and must deduce the Cartesian coordinates of every atom (Koehler & Meiler, 2011; Kuenze et al., 2019), (Schwieters et al., 2003; Wishart et al., 2008; Nerli & Sgourakis, 2019). Formally, our measurement model is $\mathbf{y} = \|\mathbf{Dx}\|_2 \in \mathbb{R}^m$, where $\mathbf{D} \in \{-1, 0, 1\}^{m \times 4N}$ is the *distance matrix* and the norm is taken row-wise (in $xyz$ space). $\mathbf{D}$ contains a single "1" and a single "-1" in each row and is not redundant (the distance between a given pair of atoms is measured at most once). $m$ corresponds to the number of measured distances.

We evaluate our method on the bromodomain-containing protein 4 (BRD4, PDB:7r5b (Warstat et al., 2023)), a protein involved in the development of a specific type of cancer (NUT midline carinoma) (French, 2010) and targeted by pharmaceutical drugs (Da Costa et al., 2013). For a given number $m$, we randomly sample $m$ pairs of alpha carbons (without redundancy) between which we assume the distances to be known. Our results are shown in Figure 4. When 500 pairwise distances or more are known, our method recovers the structural information of the target structure (RMSD < 1.77 Å, the resolution of the deposited structure resolved with X-ray crystallography). We conduct the same experiment without the diffusion model and show a drop of accuracy, highlighting the importance of the generative prior. Note that, when the diffusion model is removed, increasing the number of measurements increases the number of local minima in the objective function and can therefore hurt the reconstruction quality (plot in Fig. 4, orange curve, far-right part).

## 6    DISCUSSION

This paper introduces ADP-3D, a method to leverage a pretrained protein diffusion model for protein structure determination. ADP-3D is not tied to a specific diffusion model and allows for any data-driven denoisers to be plugged in as priors. Our method can therefore continually benefit from the development of more powerful or more specialized generative models.

Considering real data (e.g., cryo-EM, X-ray crystallography or NMR data) raises complex and exciting challenges, as experimental measurements of any one specific task typically require a lot of domain-specific processing. For example, in a real scenario, NMR experiments cannot probe long pairwise distances above 6 Å. Taking these constraints into account and applying ADP-3D to real

NMR data would be an exciting direction for future work. In cryo-EM, most of the analyzed proteins are multi-chain while the current implementation of ADP-3D only supports single-chain structures. Extending our framework to multi-chain structures, which would possible using diffusion models like Chroma, would be an impactful future direction.

In cases where the measurement process cannot be faithfully modeled due to complex nonidealities, or when the measurement process is not differentiable, our framework reaches its boundaries. Exploring the possibility of finetuning a pretrained diffusion model on paired data for conditional generation constitutes another promising avenue for future work.

## CODE AVAILABILITY

Our code is publicly available at: `https://github.com/qt7391/adp-3d-anonymous`

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
