# OpenReview forum: "Solving Inverse Problems in Protein Space Using Diffusion-Based Priors"
_ICLR.cc/2025/Conference — Submitted to ICLR 2025_

### Official Review · Reviewer_fAVV · 2024-10-31

**Soundness:** 3
**Presentation:** 2
**Contribution:** 3
**Rating:** 8
**Confidence:** 3

**Summary:**

- The authors propose to use a plug-and-play framework for diffusion models to inverse problems for determining protein structure
- This allows the authors to use off-the-shelf diffusion models such as Chroma to regularise these inverse problems
- The upshot is that this line of work is task-agnostic
- The authors showcase how their method yields promising results for three inverse problems: structure estimation from an incomplete structure, structure estimation from Cryo-EM 3D maps, and structure estimation from incomplete sparse pairwise distance data

**Strengths:**

- It is very impressive to see the results for all three inverse problems. Some aspects that are highlights to me:
    - In the structure completion it is great that the authors also test this for different levels of incompleteness. For these methods, it is good to showcase limitations to the reader
    - I am very impressed by outperforming ModelAngelo, which is as the authors mention (and to the best of my knowledge) the state-of-the-art in model building. Being able to refine the predictions to the extent in the paper is very impressive

**Weaknesses:**

- The writing can be improved upon. In particular, the introduction and the related work could really benefit from restructuring and adding additional material. This would benefit the overall flow of the paper and provide the reader with a clearer overview of the problems being tackled in the paper. Detailed suggestions below:
    - Regarding the introduction:
        - Please allow me to paraphrase:
            - Paragraph 1: Proteins can be inferred in different ways through solving an inverse problem
            - Paragraph 2: Deep learning has been key take the next step, but are limited by their way of training (new problem setting, train again to the new setting)
            - Paragraph 3: Inverse problems are dominated by learned priors
            - Paragraph 4: Diffusion models are used in the field of biology
            - Paragraph 5: Our contribution to overcome the hurdle from the second paragrap is to use diffusion models
        - I would suggest a rewrite that would look as follows (high-level)
            - The order Paragraph 1, Paragraph 3, Paragraph 2, Paragraph 5 would be more natural to me.
            - Paragraph 4 does not really serve any purpose and for me disrupts the flow of the introduction. So I would recommend removing it from the introduction and maybe move it to the background
    - Regarding related work:
        - I would expect that there would be something on finding structures from partial measurements and sparse distance matrices as well. Now there is just a lot on model building methods from Cryo-EM density maps
        - I would suggest to talk less about protein diffusion models, which seems more like something for the the background (I feel that there is not really a reason of spending a lot of time on it here)

**Questions:**

Questions for clarification
- Is the idea of using the score as a denoiser yours or is this something that is commonly used in the inverse problems field? You don’t seem to cite the works referred to in paragraph 3 of the introduction.
- On that note, what confuses me is the sentence “In particular, it is the first diffusion-based method for refining atomic models from simulated cryo-EM maps.” at the end of the abstract. Has it been used for the other two inverse problems? It would be great if the authors can be more precise here. (I assume that Cryo-EM is mentioned as it is potentially the most high-impact application, but it would be good to be clear about the other problems as well)

Additional feedback
- p.1, l.28: I would not start this paragraph with this sentence. The second sentence starting with “Experimental methods in structural biology…” would be a better choice to give the reader a better idea of what you are actually addressing in this paragraph. The first sentence can still be used later on in the paragraph to motivate the experiments.
- p.1. l.37: I would be reluctant to use the sentence “These approaches can estimate the uncertainty of their answers and provide theoretical guarantees of correctness, …” as there is little underlying theory that even guarantees that the iterates converge (the loss might converge at best, for which there is some theory in the case of RELION)
- In section 2, the authors mainly focus on ModelAngelo. I would recommend to discuss the broader problem so that the reader gets a better feeling for the broader picture
- p.2 l.90: In the second bullet of the contributions the authors claim to compare to sampling methods. In the experiment only one method is discussed. Please change this to singular (method)
- Section 4.3: It feels a bit odd that there is a whole section for the model building part from Cryo-EM maps, but that the other inverse problems are defined in the experiments. I would suggest to create a new section in between what is now section 4 and 5 and discuss all three inverse problems there. This will be useful and more straightforward for the reader who is interested in just one of the applications.
- p.7 l.360: I understand that in equation 13 several terms that are being conditioned on can be removed (\bar{x} in the last term), but for the reader this might be a big step. I would suggest to write out the steps explicitly to get from the left hand side to the right hand side
- p.8 l.409. typo: evalute -> evaluate
- p.9 l.458. What do you mean with the sentence “Note that we do not chose a structure that was resolved with cryo-EM because most cryo-EM-resolved models are incomplete.”
    - Also typo: chose -> choose

---

> ### Author Response · Authors · 2024-11-27
>
> We would like to thank our reviewer for emphasizing the quality of our results, as well as the relevance of our experiments, especially our study on the influence of the completeness level in the model refinement part. We provide an updated version of the manuscript where modifications are shown in red.
>
> ### Introduction
>
> We thank our reviewer for thoroughly reading our introduction and suggesting ways to clarify and improve it. We agree that Paragraph 4 (development of diffusion models in the space of protein atomic models) breaks the flow: it has been shortened and merged to the end of the previous paragraph. Instead of swapping Paragraphs 2 and 3, we propose to split the third paragraph in two. The first part focuses on drawing the parallel between the paths taken by protein structure determination methods and by computational imaging methods. The second part (now Paragraph 4) focuses on the possibility of using generative models as priors in inverse problems, an idea mostly introduced and developed in the field of imaging. Our introduction can now be summarized as follows:
> * Paragraph 1: Protein structure determination was first formulated as a MAP problem. However, this approach requires hand-crafted priors.
> * Paragraph 2: Fully supervised approaches have been developed and some of them redefined the state of the art. However, they need paired data and do not handle distribution shifts.
> * Paragraph 3: The field of imaging has followed the same path: MAP with heuristic priors followed by fully-supervised approaches.
> * Paragraph 4: Generative models can serve as implicit priors, which has recently been used extensively in the field of imaging. Diffusion models were developed in protein space, but have not yet been leveraged as priors for structure determination
> * Paragraph 5: ADP-3D uses a MAP framework and a prior implicitly defined by a diffusion model. It circumvents the need for heuristic priors, as well as paired data.
>
> We kindly refer to the updated version of the manuscript, where our introduction has been updated.
>
> ### Related Work
>
> The “distances-to-structure” task is a strong simplification of the reconstruction problem from NMR restraints while the cryo-EM task is more realistic. For this reason, we decided to focus on cryo-EM methods more than on NMR methods in the Related Work section. We currently provide a reference to RosettaNMR, one of the existing methods for structure determination from NMR data, in Section 5. We added references to Xplor-NIH, CS23D and CS-Rosetta.
>
> We made an effort to shorten the first paragraph of the Related Work section. Thank you for the suggestion!
>
> ### Questions
> * The idea of using the denoiser of a pretrained diffusion model in a plug-n-play framework was introduced in DiffPIR (Zhu et al., 2023), a method for solving inverse problems in image space. We mention this at the end of Section 3. To clarify this point, we added a reference to DiffPIR in the last paragraph of the Introduction.
> * ADP-3D is the first diffusion-based method applied to cryo-EM model refinement, and the first method applied to the “distances-to-structure” task. For the structure completion task, the conditioning framework introduced in the Chroma publication can be considered as an existing baseline, which we compare our results to in Figure 2. We modified the abstract to clarify this point.
>
> ### Additional feedback
> * We thank our reviewer for their suggestion regarding the first sentence of the manuscript. We rephrased it and swapped it with the next sentence.
> * Thank you for pointing out the lack of theory on convergence of existing MLE and MAP-based reconstruction methods. We rephrased the last sentence of the first paragraph.
> * We de-emphasized the focus on ModelAngelo in the second paragraph of the introduction.
> * We corrected the second contribution: “We outperform an existing posterior sampling method at reconstructing full protein structures from partial structures”.
> * Thank you for your suggestion regarding Section 4.3. Although it would be possible to have an additional section to describe the three forward models, the first and third tasks can be compactly described with one equation. In the interest of conciseness, we chose to briefly describe the forward model of each task at the beginning of each paragraph of Section 5. For the model refinement task, the forward model is more complex and we describe the technical contributions that are specific to this task in Section 4.3 (e.g., MC sampling of the side-chain angles).
> * Thank you for the suggestion, we added an intermediate step in Equation 13.
> * For the model refinement task, we moved our results to the supplements and replaced them with results obtained on real data. We removed the unclear remark highlighted by our reviewer.

---

> > ### Author Response · Authors · 2024-12-03
> > **Review Editing Deadline**
> >
> > Dear reviewer, the time window for editing reviews is closing tonight. We have made every effort to address your concerns, please let us know if you have any further questions. Thank you for your thoughtful engagement with our work.

---

> > > ### Comment · Reviewer_fAVV · 2024-12-03
> > >
> > > Thanks to the authors for the responses! I am happy with the changes and feel that the current score is still an adequate representation of the work.

---

### Official Review · Reviewer_1GSm · 2024-11-03

**Soundness:** 3
**Presentation:** 2
**Contribution:** 2
**Rating:** 3
**Confidence:** 4

**Summary:**

The paper introduces ADP-3D (Atomic Denoising Prior for 3D reconstruction), an algorithm to combine a pretrained protein diffusion model with additional data. The method uses a plug-n-play approach that splits the log posterior into a prior and likelihood term, duplicates the variables and couples them via a quadratic term. This allows the authors to avoid incorporating the likelihood directly into the diffusion process. Instead only a simple quadratic term needs to be added to the likelihood, and a diffusion step is used to incorporate the prior. No retraining of the diffusion model is necessary. The authors focus on the Chroma model (Ingraham et al., 2023) for protein structures and combine the model with various likelihoods (structure completion from sub-sampled backbone positions, model refinement based on a density map, sequence information, incomplete distance matrix). Using specific examples, the authors find that the incorporation of the diffusion process improves the quality of the reconstructed structures.

**Strengths:**

* A general approach to combining pretrained diffusion models with data based on a variable-splitting framework.
* The resulting algorithm is quite simple and allows for combining pretrained diffusion models with new data (thereby avoiding a retraining step).
* The approach is illustrated for various protein structure modeling tasks.

**Weaknesses:**

* The general approach (algorithm 1) seems to be a minor modification of existing work.
* The method is illustrated only for simulated data.

__Recommendation__

I recommend to reject the article. ADP-3D is a straight forward application of the plug-n-play framework to protein diffusion models. The idea of using a variable splitting approach to combine diffusion models with additional data has already been proposed by Zhu et al. (2023) in the context of image restoration. So the major novelty is in the application of the method to protein structure modeling. However, the authors only use simulated data in their numerical experiments.

**Questions:**

* The abstract states that this is "the first diffusion-based method for refining atomic models from simulated cryo-EM maps." Why did you not apply the method to real cryo-EM maps? The paper would be much stronger, if you showed applications to real data. Likewise, you could use NMR distance bounds as experimental distance information rather than simulated distances.
* How do you choose the parameter $\rho$? Are the results sensitive to the choice of $\rho$?
* How do you define __completeness__? You state that completeness is the "RMSD of the predicted structure for the X% most well-resolved alpha carbons (compared to the deposited structure)." How do you assess if an alpha carbon is well-resolved? Please note that the resolution of your model (in my understanding something like its precision) and its accuracy are not the same thing.
* Figures 3, 4, 5 report the resolution of the deposited structure. What is the purpose of showing this number? The resolution of the PDB structure does not tell us anything about the difficulty of the structure calculation task.
* Figure 5, right panel: Why does the RMSD increase with increasing number of distances (#pd > 4k) when not using a diffusion model? It should go down and should also approach zero for as many distances as 8 k pairwise distances for a protein of length 127 residues.

__Additional feedback__

* The selection of distances is unrealistic in the sense that, for example, NMR experiments such as NOESY only probe short distances up to 6 \AA. Since there are many more long-range distances, your way of selecting distances randomly will tend to pick distances that are not experimentally accessible, which makes the selection process unrealistic.
* The abstract states that "raw biophysical measurements such as cryo-EM density maps" can be used by your method. To categorize cryo-EM maps as raw data is misleading. A density map is the result of a complex sequence of processing steps starting with motion correction, particle picking, etc. The raw measurements in cryo-EM are the micrographs.
* Figure 1 does not add much information beyond what is already said in the text. You could remove the figure and use the space to show more details about your numerical experiments such as your tests with RFdiffusion (Watson et al., 2023).
* Likewise, figure 2 is a specific result for the correlation matrix used by Chroma. It could be moved to the supplementary information.
* On page 8 you state that you avoid testing on structures that were part of the Chroma training set by using only structures that were published after the release of Chroma. However, this does not guarantee that the Chroma training set and your test structure do not overlap. First, many examples of structures of one and the same protein (or a protein with a highly similar sequence) can be found in the PDB. Second, protein structures are much more persistent than sequences. So even proteins with a low sequence similarity can have similar structures.
* In the captions of figure 3 and 5, you report the lowest RMSD obtained with 8 runs. Please also report the maximum RMSD (you could also report the mean and the standard deviation) to give an indication of how much your models vary.
* There is a slight inconsistency in your whole approach: You claim to do MAP estimation, but then you use diffusion-based sampling.

---

> ### Author Response · Authors · 2024-11-27
> **Response (1)**
>
> We thank our reviewer for their detailed comments and feedback. We provide an updated version of the manuscript where modifications are shown in red. In this revised manuscript, we have now added results on experimental cryo-EM density maps for the model refinement task. We respectfully disagree with the characterization of ADP-3D as a straightforward adaptation of prior diffusion modeling approaches. While our work builds on the general plug-and-play framework, we believe the application to protein structure modeling is a substantial contribution in itself. Furthermore, ADP-3D introduces several domain-specific technical innovations that are critical for its success and go beyond a direct translation of existing methods.
>
> ### Main contribution
>
> As highlighted by our reviewer, the main contribution of this paper is the introduction of a method leveraging any pretrained diffusion model for solving a large spectrum of inverse problems in protein space. The method is inspired by DiffPIR (Zhu et al., 2023), a plug-n-play method designed for image inpainting, deblurring and super-resolution. ADP-3D operates in protein space and is evaluated on three different tasks using two different diffusion models of protein atomic models. We argue that ADP-3D is not a straightforward translation of previous work. Several technical contributions are specific to the fact that we are operating in the space of atomic models. For example, the preconditioning strategy (Section 4.2 and Appendix A) used in the structure completion and model refinement tasks is key for convergence and represents one of the methodological contributions of this paper (see, for example, our ablation study in Figure S4). For the model refinement task, our coarse-to-fine (i.e., frequency marching) strategy is also necessary for convergence and only applies to this task.
>
> We would like to emphasize the fact that the paper demonstrates the versatility of the method and the possibility of using it as a true “plug-n-play” method. ADP-3D can be used to leverage any pretrained diffusion model as a prior, which we demonstrate by using both Chroma and RFdiffusion. ADP-3D is compatible with any inverse problem for which the likelihood of the measurements is differentiable with respect to the atomic model, making it a very general tool, which we demonstrate by tackling three different problems: structure completion, model refinement and distances-to-structure.
>
> ### Results with experimental data
>
> In order to demonstrate the applicability of ADP-3D to real scenarios, we provide additional results using experimental density maps. Our results are described in Section 5 (Figure 3) and replace our previous results on synthetic data (moved to the Supplementary Materials, G.6). We use three publicly available density maps of single-chain structures: EMD-33209 (map for PDB:7xkw), EMD-32760 (map for PDB:7wsm) and EMD-26482 (map for PDB:7ug0). We show that, by leveraging the pretrained model Chroma as a prior, ADP-3D can refine incomplete models provided by ModelAngelo. These results validate the applicability of the method on experimental data, thereby proving its relevance in real scenarios. We kindly refer to the updated Supplementary Materials for further details.
>
> ### Influence of $\rho$
>
> We realize that our notation is overloaded and apologize for the confusion. $\rho$ represents both the penalty parameter of a proximal operator and the “momentum” of a gradient step. We assume that our reviewer’s question concerns the former.
>
> In ADP-3D, we do not perform a full maximization on the proximal operator. Instead, we perform a single gradient step (with momentum) from the iterate $\tilde{z}_0$ (L259, page 5). In $\tilde{z}_0$, the gradient of the penalty term $(\rho/2)\Vert z-\tilde{z}_0\Vert_2^2$ with respect to $z$ is $0$ in $\tilde{z}_0$. Therefore, $\rho$ has no influence on the results. We now explicitly mention this at the end of Section 4.1 and use a different notation ($\gamma$) for the penalty parameter to avoid the potential confusion with the momentum parameter. We thank our reviewer for raising this point.
>
> ### Clarification on “completeness” vs. RMSD
>
> We define the “completeness” of an incomplete model as the ratio between the number of  modeled residues over the total number of residues. This term is sometimes replaced with “completion”, like in the ModelAngelo publication. When plotting completeness vs. RMSD, we first align the fully reconstructed atomic model with the deposited model. We then compute the Euclidean distances between the two models, for each alpha-carbon. The alpha-carbons are ranked by increasing deviation to the “ground truth” location. We then compute the RMSD between the two models, for the $N$ first alpha-carbons only, where $N$ ranges from 1 to the total number of residues. This gives $N$ points in the “completeness vs. RMSD” plot.

---

> > ### Author Response · Authors · 2024-11-27
> > **Response (2)**
> >
> > ### Purpose of showing deposited resolution
> >
> > We agree with our reviewer that the resolution of the deposited PDB structure does not indicate the complexity of the task. The conditioning information (incomplete model for structure completion, density map for model refinement) is directly derived (i.e., simulated) from the PDB structure and the latter can therefore be considered as a “ground truth” structure, independently of the deposited resolution. However, this is not the true structure, meaning that the prior could potentially bias the reconstruction towards a structure that is different from the deposited structure (considered here as ground truth) and closer to the true structure. In practice, this means that the RMSD to the “ground truth” structure is different from the RMSD to the true structure. The magnitude of the difference is on the order of the deposited resolution.
> >
> > ### Increasing resolution with number of pairwise distances
> >
> > We hypothesize that the increase in RMSD with the number of pairwise distances (Figure 4, orange curve, far-right part) comes from the fact that the problem is not convex and the number of local minima increases with the number of distances. Gradient-based optimization is more likely to get stuck in a suboptimal state when the number of local minima is high. This hypothesis is corroborated by the decrease in minimum RMSD and the increase in variance. This is mentioned on page 9: “Note that, when the diffusion model is removed, increasing the number of measurements increases the number of local minima in the objective function and can therefore hurt the reconstruction quality.”
> >
> > ### Additional feedback
> >
> > **Long range interactions in NMR.** We thank our reviewer for providing additional details on the experimental constraints of NMR. We agree that the “distances-to-structure” task is a strong simplification of the reconstruction problem in NMR and is only intended as a loose analogy (L458, page 9). We now explicitly mention the impossibility of probing long distances (> 6A) in a real scenario in the Discussion section.
> >
> > **Terminology.** We agree with our reviewer that the term “raw” can be misleading for cryo-EM data. We removed it.
> > Figures. Following our reviewer’s suggestion, we moved Figure 2 to the Supplementary Materials. Figure 1 does not add information to the text, but currently serves as a “teaser” figure that could help the reader rapidly understand how the method works at what it is capable of.
> >
> > **Choice of target structures.** As mentioned by our reviewer, choosing a protein that was released on the PDB after the most recently deposited structure used to train Chroma is not sufficient to ensure this is a “new” structure for the model. For this reason, the majority of the structures used in the supplements belong to the CASP15 dataset (see Table S1).
> >
> > **Variability of the output.** To indicate the variability of the output of the method, we currently show in Table S3 the number of outputs that end up with an RMSD below 1.5A. For the model refinement task, the variability of the output is quantified in Figure S6. While some of the runs fail to reach sub-2A RMSD, we show that these outliers can be removed by measuring how well the structures fit into the input density map (using the log-likelihood function). This outlier removal process can therefore operate without access to ground truth information, making it straightforward to apply on new molecules. We thank our reviewer for the suggestion regarding Figures 3 and 5 and will make an effort to further indicate the maximum RMSD obtained over 8 runs.
> >
> > **MAP vs. posterior sampling.** We use a diffusion model and our output is stochastic (random initialization and random sampling step in Algorithm 1), but our method is not a posterior sampling method. As mentioned in Zhu et al. (2023) [1], the method is inspired by the plug-n-play framework and is better described as a MAP estimation method. Importantly, the requirements for a convergence guarantee do not hold in general, hence the variability of the output (due to the dependence on initialization) in plug-n-play reconstruction methods.
> >
> > [1] Zhu, Yuanzhi, et al. "Denoising diffusion models for plug-and-play image restoration." Proceedings of the IEEE/CVF Conference on Computer Vision and Pattern Recognition. 2023.

---

> > > ### Author Response · Authors · 2024-12-03
> > > **Review Editing Deadline**
> > >
> > > Dear reviewer, the time window for editing reviews is closing tonight. We have made every effort to address your concerns, please let us know if you have any further questions. Thank you for your thoughtful engagement with our work.

---

### Official Review · Reviewer_gXXw · 2024-11-04

**Soundness:** 2
**Presentation:** 3
**Contribution:** 3
**Rating:** 5
**Confidence:** 4

**Summary:**

The paper introduced a framework, ADP-3D, which uses a diffusion model as a prior to solve the inverse problem in structural biology. The authors leverage the pretrained diffusion models in the atomic model space like Chroma and RFDiffusion as their prior. The downstream tasks include structure completion (as a toy problem), atomic model refinement (model completion given a simulated density map, an incomplete atomic model, and the sequence), and solving structure from given pairwise distances.

**Strengths:**

As far as my knowledge goes, this work is one of the first to leverage a pretrained diffusion model of protein atomic structures as a Bayes prior, to solve the inverse problems which are very common in structural biology. The connection between the generative model and the experimental observation is very important in expanding the scope of the AI for science field.

**Weaknesses:**

1. Since the paper uses pretrained diffusion models from Chroma and RFDiffusion, and the measurement models for the tasks are apparent, from a methodological standpoint, my understanding is that the main contribution of this paper is the MAP estimation method given a diffusion prior. As ADP-3D seems like a generic algorithm that is not heavily tailored for structural biology, there are many similar methods in the field of diffusion posterior sampling for inverse problems, e.g. DPS, $\Pi$GDM, as surveyed in [1] and also mentioned in the related work section by the authors. The authors should compare the similarities and differences between the propose ADP-3D method and other similar algorithms. The authors should also use at least one of these methods as the baseline and compare the results to show the advantage of the proposed ADP-3D method.

2. I understand that real data is not considered in the scope of this paper, but this makes the downstream tasks a bit far from the real world implementation. For example, the model refinement task uses a small, simulated density as a given condition, while ModelAngelo itself was trained on real experimental density maps. In reality, it is possible that ModelAngelo does not even output a severely incomplete model, making the whole setting of this task futile.

3. The model refinement task seems to only contain one example (7pzt).


[1] Daras, Giannis, et al. "A survey on diffusion models for inverse problems." arXiv preprint arXiv:2410.00083 (2024).

**Questions:**

1. How does ADP-3D related and differ from other diffusion posterior sampling methods?

2. In the model refinement task, what happens if using an experimental density map as the condition? The authors claimed that "most cryo-EM-resolved models are incomplete", but in my opinion the authors should at least have one more realistic case to show the setting in this task is not a castle in the air.

---

> ### Author Response · Authors · 2024-11-27
>
> We thank our reviewer for highlighting the novelty of our method leveraging a pretrained diffusion model of protein atomic structures as a Bayesian prior, and its role in establishing new connections between generative modeling and scientific disciplines. We also thank the reviewer for their feedback and suggestions. We provide an updated version of the manuscript where modifications are shown in red. Our main additions are:
> * Additional experiments with real data for the model refinement task.
> * A comparison to the DPS method for the structure completion task.
>
> ### Additional results with experimental data
>
> In order to demonstrate the applicability of ADP-3D to real scenarios, we provide additional results using experimental density maps. Our results are described in Section 5 (Figure 3) and replace our previous results on synthetic data (moved to the Supplementary Materials, G.6). We use three publicly available density maps of single-chain structures: EMD-33209 (map for PDB:7xkw), EMD-32760 (map for PDB:7wsm) and EMD-26482 (map for PDB:7ug0). We show that, by leveraging the pretrained model Chroma as a prior, ADP-3D can refine incomplete models provided by ModelAngelo. These results validate the applicability of the method to experimental data, thereby proving its relevance in real scenarios. We kindly refer to the updated manuscript and Supplementary Materials for further details.
>
> ### Comparison to other diffusion-based posterior sampling methods
>
> Thank you for raising this point. As mentioned by our reviewer, there exist many different methods to perform posterior sampling using diffusion models in image space (DPS [1], PiGDM [2], etc…). We also thank our reviewer for pointing us to the survey [3], which we added to the second paragraph of the Related Work section in addition to other related works.
>
> While designing the method, we experimented with a few of these techniques, in particular with DPS, ScoreALD [4] and DiffPIR [5]. DiffPIR is generally more accurate than scoreALD because gradients of the log-likelihood are computed on the denoised, protein-like structure. DPS uses the same idea, but gradients are computed through the denoiser, making the whole computation significantly slower.
>
> Following our reviewer’s suggestion, we now include a comparison to DPS in the Supplementary Materials (Appendix G.4 and Figure S3). We use the structure completion task on PDB:8ok3 with a subsampling factor of 4. For DPS, we run a sweep over the parameter $\zeta^\prime$. Both methods run over 1,000 steps on 64 replicas. We find that ADP-3D leads to more accurate results and is, on average, about six times faster than DPS because it does not need to compute gradients through the denoiser. We kindly refer to the updated Supplementary Materials for further details.
>
>
> [1] Chung, Hyungjin, et al. "Diffusion posterior sampling for general noisy inverse problems." arXiv preprint arXiv:2209.14687 (2022).
>
> [2] Song, Jiaming, et al. "Pseudoinverse-guided diffusion models for inverse problems." International Conference on Learning Representations. 2023.
>
> [3] Daras, Giannis, et al. "A survey on diffusion models for inverse problems." arXiv preprint arXiv:2410.00083 (2024)
>
> [4] Kawar, Bahjat, et al. "Denoising diffusion restoration models." Advances in Neural Information Processing Systems 35 (2022): 23593-23606.
>
> [5] Zhu, Yuanzhi, et al. "Denoising diffusion models for plug-and-play image restoration." Proceedings of the IEEE/CVF Conference on Computer Vision and Pattern Recognition. 2023.
>
> ### ModelAngelo on synthetic data
>
> We agree with our reviewer that, since ModelAngelo was trained on experimental maps but is here used on simulated maps, the input may be out-of-distribution and ModelAngelo’s output more incomplete than in a real setup. The completeness we obtained with ModelAngelo on PDB:7pzt ranges between 50% and 70% (Figure S5b). In comparison, the completeness obtained with ModelAngelo on experimental maps ranges between 16.3% and 100% (mean $\pm$ std = 80 $\pm$ 24%, see Extended Data Table 1 in [6]), These numbers show that ModelAngelo provides (sometimes significantly) incomplete models and motivate the need for a refinement method.
>
> [6] Jamali, Kiarash, et al. "Automated model building and protein identification in cryo-EM maps." Nature 628.8007 (2024): 450-457.

---

> > ### Comment · Reviewer_gXXw · 2024-12-02
> >
> > I thank the authors for the response and the added experiments, especially on the real cases of the atomic model refinement task. However, for the first two cases (EMD-33209, EMD-32760), besides some loop completion, ADP-3D also made significant modifications on the original models by making the top right helix longer and adding a new helix to the original model. The new modified atomic models does not seem to agree with the experimental densities very well, which is actually quite concerning. Maybe a model-to-map FSC vs densities comparing MA and ADP-3D could show how convincing the "refinement" results are. As pointed out by the other reviewer, the setting for the pairwise distance to structure task is also unrealistic compared to the setting in a real NMR experiment. While I strongly believe that this paper is a step forward to using pre-trained diffusion models to solve problems in the protein atomic model space, the novelty of the proposed algorithm seems limited from a purely methodology stand point, while the applications are not very convincing. I decide to keep my score.

---

### Author Response · Authors · 2024-11-28
**Shared Response**

We sincerely thank all our reviewers for their constructive feedback. We value their appreciation of the novelty of our approach **[gXXw]**, the quality of our results **[fAVV]** and our illustration of the method on diverse protein structure determination tasks **[1GSm]**. We are glad they emphasized the significance of this work, establishing new connections between generative modeling and scientific disciplines **[gXXw]**. We appreciate their positive comments on the relevance of our experiments **[fAVV]**.

We provide an updated version of the manuscript where modifications are shown in red. Our main modifications include:
* The addition of results on experimental cryo-EM structures for the model refinement task instead of only synthetic data.
* A comparison to the Diffusion Posterior Sampling (DPS) method.
* A re-wording in the Introduction and Related Work sections.

In order to clarify our contribution, we would like to emphasize that ADP-3D demonstrates, for the first time, the possibility of using a pretrained diffusion model as a Bayesian prior for protein structure determination **[gXXw]**. While our approach builds on recent works solving inverse problems in image space (e.g., inpainting, deblurring, super-resolution), we believe the application to protein structure modeling is a substantial contribution in itself. To this end, ADP-3D introduces several domain-specific technical innovations that are critical for its success and go beyond a direct translation of existing methods. The paper demonstrates the versatility of the method by tackling three different inverse problems, and the possibility to leverage any user-chosen diffusion model as a prior.

In order to enable the scientific community to use and build on ADP-3D, we have recently made our code publicly available and fully open source on github: https://github.com/qt7391/adp-3d-anonymous

---

### Meta-Review · Area_Chair_L4RX · 2024-12-20

**Metareview:**

The authors address inverse problems in structural biology by using pre-trained diffusion modeling. The paper received three reviews with two arguing for rejection and one for acceptance. In general, the reviewers appreciated the application of SOTA AI models to fundamental scientific problems. However, concerns were raised that it represents a straightforward modification of existing work and only considers simulated data. Two reviewers agreed that, in their opinions, the distances chosen to evaluate the method are not appropriate for the NMF experiment. Overall, the sense was that the technical AI/ML contribution of this work did not rise to what would be expected for ICLR.

**Additional Comments On Reviewer Discussion:**

One reviewer closely considered the rebuttal and the comments of the other reviewers and agreed with the negative review that the paper did not reach the threshold for acceptance. The issues with the technical novelty and experimental evaluation were too significant to be left to a final revision, but should be done before resubmission elsewhere.

---

### Decision · Program_Chairs · 2025-01-22

Reject